# Evaluating TNF-α and Interleukin-2 (IL-2) Levels in African American Primary Open-Angle Glaucoma Patients

**DOI:** 10.3390/genes13010054

**Published:** 2021-12-25

**Authors:** Teja Alapati, Kyra M. Sagal, Harini V. Gudiseva, Maxwell Pistilli, Mark Pyfer, Venkata Ramana Murthy Chavali, Joan M. O’Brien

**Affiliations:** Scheie Eye Institute, Department of Ophthalmology, University of Pennsylvania, Philadelphia, PA 19104, USA; Tejavu42@gmail.com (T.A.); kmsagal@gmail.com (K.M.S.); gudiseva@pennmedicine.upenn.edu (H.V.G.); pistilli@pennmedicine.upenn.edu (M.P.); Mark.pyfer@pennmedicine.upenn.edu (M.P.); vchavali@pennmedicine.upenn.edu (V.R.M.C.)

**Keywords:** SNP genotyping, gene association, GWAS SNP, TNF-α, IL-2, plasma biomarker, endophenotypes, IOP, African American, POAG, sandwich ELISA, cytokines

## Abstract

Purpose: To establish if SNPs in *TNF-α* and *IL-2* genes are associated with Primary Open-Angle Glaucoma (POAG) in African Americans (AA). We also determined whether plasma TNF-α and IL-2 levels could serve as biomarkers for POAG in African Americans using sandwich enzyme-linked immunosorbent assay. Methods: A single SNP association analysis was performed to investigate the association between potential gene variants in *TNF-α* and *IL-2* genes and POAG in the AA population. Plasma samples from 190 African Americans (72 from normal subjects and 118 POAG cases) were obtained for TNF- α studies and 367 samples (135 from normal subjects and 232 from POAG cases) were obtained for IL-2 studies. TNF-α levels and IL-2 levels were measured by sandwich enzyme-linked immunosorbent assays (ELISA) and analyzed to see if they reached significance in cases with POAG and endophenotypes when compared to normal subjects. Results: The SNP, rs1800630, in TNF-α gene was found to be marginally associated with POAG. SNPs in *IL-2* gene were not associated with POAG in the case-control analysis. No significant difference was found between TNF-α levels and IL-2 levels in normal and POAG case subjects in our study. IL-2 levels were inversely correlated with high IOP in POAG cases. Conclusions: Although we found a marginal SNP association of TNF-α, assessing the expression levels of TNF-α and IL-2 may serve as promising biomarkers for African American POAG. Further investigation is needed to determine if POAG can be subdivided into more specified cohorts of the disease, which may affect plasma cytokine levels differently.

## 1. Introduction

Glaucoma is a group of optic neuropathies consisting of progressive damage to retinal ganglion cells (RGCs) and their axons, often leading to loss of peripheral vision and eventual blindness [1]. Primary open-angle glaucoma (POAG), which accounts for more than 70% of all glaucoma subtypes [2], is a chronic, slow degeneration of the optic nerve, clinically characterized by an obstructed anterior chamber iridocorneal angle [3], often resulting in an increased intraocular pressure (IOP) [4]. POAG is the second leading cause of blindness worldwide after cataracts [2] and is the leading cause of blindness in African Americans [1]. Approximately 3.4 million Americans are expected to have POAG by 2020, with over half being undiagnosed [5].

POAG is clinically well-defined, but the precise pathological mechanism underlying the glaucomatous damage is not fully understood. This disease is multifactorial, and well-established risk factors include male gender, African ancestry, genetic predisposition, thinner central corneal thickness, and older age [6]. However, elevated IOP remains the only modifiable risk factor. Nonetheless, a third of POAG cases exhibit normal IOP, and a sizable proportion have progressive disease even if they have a therapeutically lowered IOP [7]. Other risk factors, including neurotoxicity [8] and immunologic factors [9], may also play a role in the pathogenesis of POAG. Advanced screening methods to diagnose early POAG are lacking in efficacy, particularly when based on IOP, since a subset of newly diagnosed POAG patients exhibit normal IOP [10]. Given this poor sensitivity, novel biomarkers of early POAG are warranted.

Biomarker panels exist for other common, multifactorial conditions such as cardiovascular [11] and Alzheimer’s disease [12], allowing for improved diagnostics and management. In addition to increased IOP, other well-investigated factors associated with glaucoma include a decreased ocular blood flow, ocular vasculature dysregulation [7], blood pressure changes, [13] oxidative stress, and inflammation [14]. There is evidence of significant oxidative damage driving the loss of trabecular meshwork (TM) cells in patients with POAG [15]. The role of cytokines in oxidative stress and inflammation contributing to glaucomatous degeneration is of particular importance. Immunological abnormalities could affect the pathophysiology of RGC apoptosis. Prior investigations showed varying levels of immune factors in the blood of individuals with POAG [16]. Cytokines, such as interleukins, interferon-g and tumor necrosis factor alpha (TNF-α), have been investigated and were found to be associated with POAG [9,17].

The chief source of cytokines, CD4+ T-helper cells, are divided into two classes based on their secretion profiles. T-helper 1 (Th1) cells regulate cellular immunity by producing interleukin (IL)-2, IL-12p70, IL-23, interferon-gamma, and TNF-α. T-helper 2 (Th2) cells contribute to humoral immunity via the secretion of IL-4, IL-5, IL-6, IL-10, and IL-13. The balance of Th1-derived and Th2-derived cytokines plays an important role in immunoregulation, evidenced by an imbalance of Th1/Th2 cytokines associated with many diseases [18,19]. Data supporting the role of immune balance in the mechanism of glaucomatous neuropathy also exist [20]. IL-4, IL-6, and IL-10 have been shown to have a protective role in RGCs in POAG [21,22]. IL-2, IL-5, IL-10, and TNF-α are known to mediate a host of inflammatory pathways [23,24]. IL-2 induces T-cell proliferation and regulates cytotoxic T and regulatory T cells in addition to antibody production. Elevated levels of IL-2 mRNA have been found on the irises of patients with neovascular glaucoma [25].

Tumor necrosis factor alpha (TNF-α) is a pro-inflammatory cytokine known to primarily induce RGC death through mechanisms including mitochondrial dysfunction, oxidative stress, and receptor-mediated caspase activation [26,27]. Studies showing increased TNF-α in the aqueous humor and serum of POAG patients [28,29,30] suggest that TNF-α may be a potential POAG biomarker, and TNF-α levels could be a marker of glaucoma severity. Increased levels of TNF-α were found in the plasma of individuals who experienced trauma, and these increased levels were correlated with the severity of injury [31]. It can thus be hypothesized that POAG patients would have higher TNF-α plasma levels with increased amounts of RGC death. While increased levels of TNF-α were previously found in the plasma of a Saudi Arabian population [30], it is important to compare plasma TNF-α levels of POAG cases in an African American population to better understand the disease in this demographic.

To determine the role of oxidative stress and inflammation in POAG, we investigated the relationship between plasma IL-2, TNF-α levels with POAG and its endophenotypes. Our research group performed a large-scale GWAS project titled the Primary Open-Angle African American Glaucoma Genetics (POAAGG) study. Case-control analyses and single SNP association analyses were performed to investigate the association of potential gene variants in *TNF-α* and *IL-2* genes with POAG in the AA population [32]. We aimed to understand the pathogenesis of POAG for both diagnostic and potential therapeutic purposes and investigate if IL-2 and TNF-α levels are affected in plasma samples of African Americans diagnosed with POAG compared to normal healthy subjects. We will determine if altered levels could serve as a biomarker to detect early glaucoma and/or POAG. Quantifying IL-2 and TNF-α in glaucoma patients may additionally help monitor disease progression and the efficacy of therapy.

## 2. Materials and Methods

### 2.1. Study Population and Criteria

The study population was composed of self-identified African Americans recruited for the Primary Open-Angle African American Glaucoma Genetics (POAAGG) study [32]. In brief, inclusion criteria included age over 35 years and self-identification as Black (African-American, African descent, or African-Caribbean). Exclusion criteria encompassed history of secondary causes of glaucoma: narrow angle, closed-angle, neovascular, mixed-mechanism, pseudoexfoliation, surgery or trauma-induced. Patients were also excluded if they had any history of uveitis, Grave’s disease with ocular manifestations, iris neovascularization, optic nerve atrophy from other diagnoses, or advanced proliferative diabetic retinopathy. A comprehensive eye exam included IOP measurement, gonioscopy, dilated fundus and optic disc examination, visual fields, stereo disc photography, optical coherence tomography imaging, and measurement of central corneal thickness. POAG cases were identified by fellowship-trained glaucoma specialists as having an open iridocorneal angle, characteristic visual field defects in at least one eye on two consecutive field tests, and at least one of the following optic nerve findings: excavation, neuroretinal rim thinning, nerve fiber layer deficit, or notching. Normal controls were defined as subjects older than 35 lacking the following: high myopia or presbyopia, family history of POAG, abnormal visual fields, IOP greater than 21 mmHg, characteristic glaucomatous optic nerve findings, optic nerve symmetry, and cup-to-disc ratio difference between eyes greater than 0.2.

Informed consent was obtained from all participating subjects, and the research followed the Institutional Review Board of the University of Pennsylvania and the Declaration of Helsinki for experiments involving human tissue. Based on sample availability, a case-control cohort comprising 232 POAG cases and 135 control subjects was used to study IL-2 levels, and 118 POAG cases and 72 control subjects were used for TNF-α studies. Subjects who withdrew or opted out of participation in our study were excluded. Samples for both studies were selected based on the serum availability and were randomly selected for our study.

### 2.2. Genotyping

Blood/Saliva samples were collected from the above POAAGG population. Genotyping and quality control analysis were performed as previously described [32]. DNA was isolated from blood using Gentra PureGene kits (Qiagen, Valencia, CA, USA). Saliva was collected from subjects via Oragene DISCOVER (ORG-500) self-collection kits (DNA, Genotek, Ottawa, ON, Canada). Genomic DNA was isolated and sequenced from either blood and/or saliva samples. DNA concentrations were confirmed with the fluorescence-based Quant iT dsDNA Board-Range Assay kit (cat # Q33130, Life Technologies, Carlsbad, CA, USA), following manufacturer’s instructions. DNA samples from cases were successfully genotyped using the Multi-Ethnic Genotyping Array (MEGA) V2 (EX) consortium chip on the Infinium iSelect platform (Illumina, San Diego, CA, USA). The SNPs near *TNF-α* and *IL-2*, including upstream and downstream regions containing 5′ and 3′ untranslated regions, were extracted from the genotyping data. A case-control association analysis was performed as described in POAAGG GWAS study [32]. Single variant, binary association tests were performed genome-wide using a logistic regression model framework, as implemented in the PLATO software package and described in POAAGG GWAS study [32].

### 2.3. Sample Collection

Blood samples from patients were collected by venipuncture in BD Vacutainer Cell Preparation Tubes (CPT) containing Sodium Heparin anticoagulant (Cat #362753) ollowing standard techniques and manufacturer instructions (BD Biosciences, Franklin Lakes, NJ, USA). The sample was then gently remixed via inversion prior to centrifugation for 15 min at 3000 rpm. Following centrifugation, the plasma layer at the top was transferred into a new 15 mL conical tube. This plasma was aliquoted in 1 mL volumes into 1.5 mL centrifuge tubes, which were labeled and stored at −80 °C.

### 2.4. ELISA to Assay IL-2 and TNF-α

Plasma IL-2 levels were measured using a commercial ELISA kit (Biolegend ELISA MAX™ Standard Set Human IL-2 Cat. #431801, Becton, Dickinson and Company, Franklin Lakes, NJ, USA). The TNF-α were measured using human TNF-α DuoSet ELISA kit (Cat# DY210, Biotechne, Minneapolis, MN, USA), following the manufacturer’s instructions. In brief, the capture antibody for both the assays was diluted in coating buffer to working concentration, and 100 µL of the diluted capture antibody was added to each well. The plate was sealed and incubated overnight at room temperature. A standard curve was used for each assay, and the standards ranging from 3.9 to 500 pg/mL were prepared using two-fold serial dilution. Assay Diluent was used as the zero standard (0 pg/mL) and negative control. Standards, samples, and controls were all run in duplicate on each plate. The plate was washed four times with at least 300 µL of Wash Buffer (Biolegend Cat. #421601, Biolegend, San Diego, CA, USA). Residual buffer was blotted by firmly tapping the plate upside down on an absorbent paper. All subsequent washing steps were performed similarly.

To block non-specific binding and reduce background, 200 μL Assay Diluent (Biolegend Cat #421203, Biolegend, San Diego, CA, USA) with 1X PBS containing bovine serum was added to each well. The plate was then sealed with an adhesive cover and incubated at room temperature for 1 h with gentle shaking on a rotary shaker. After washing, 100 µL of standards and samples were added in duplicate, and the plate was incubated with gentle shaking for two hours. The plate was washed, and 100 μL of detection antibody, diluted in Assay Diluent, was added to each well. The plate was washed after 2 h. Next, 100 µL of Avidin-HRP solution diluted in assay diluent was added per well, followed by a 30-min incubation. For the final wash, the plate was washed 5 times, with a brief 30 s incubation per wash to help minimize background. Then, 100 µL of TMB Substrate Solution (BioLegend Cat. #421101, Biolegend, San Diego, CA, USA) was added to each well and incubated in the dark for 20 min. The reaction was stopped by adding 100 µL of stop solution (BioLegend No. 423001, Biolegend, San Diego, CA, USA) to each well. Within 15 min, plate absorbance was measured at 450 nm and 570 nm using a TECAN Infinite M200 Pro instrument. The optical density values obtained at 570 nm were subtracted from those obtained at 450 nm to correct for optical imperfections. A 2nd-order polynomial curve-fitting algorithm was used to interpolate the standard curve and calculate sample TNF-α and IL-2 concentrations.

#### TNF-α and IL-2 Concentration Normalization

Total protein concentration was used to normalize TNF-α and IL-2 concentrations by dividing TNF-α and IL-2 (pg/mL) by total protein concentration (mg/mL). Total protein concentration was determined using a Qubit Protein Assay (ThermoFisher, Waltham, MA, USA; Cat. No. Q33211). Samples were diluted 1:20 and the assay was performed as per manufacturer’s instructions. Samples were read on a microplate reader by diluting the three standards to 20 µg/mL, 15 µg/mL, 10 µg/mL, and 0 µg/mL. Fluorescence was read at excitation/emission maxima of 470/570 nm.

### 2.5. Statistical Analysis

Data were presented as mean ± standard deviation for TNF-α, IL-2 levels and other clinical variables. Analyses were performed using a *t*-test (two tailed) and chi-square test methods to evaluate the differences between the two study groups. Data were log transformed to obtain a more normal distribution for reliable statistical analysis. An unpaired *t*-test on log-transformed data for TNF-α, IL-2 concentration (pg/mL) was performed to determine if the difference was significant. A Spearman rho test with cluster sampled bootstrap correlations was performed to test the association of log-transformed IL-2 with age and several endophenotypes, and linear regression, with general estimating equations to account for the correlation between eyes, was used to calculate p-values for the endophenotypes where both eyes were used for each study participant [33]. *p*-values of less than 0.05 were considered statistically significant. Statistical analysis was performed with GraphPad Prism 8 (GraphPad Software, Inc., San Diego, CA, USA) and SAS 9.4 (SAS, Cary, NC, USA).

## 3. Results

### 3.1. Genotyping Analysis

Association studies showed minimal associations for SNP r1800630 in *TNF-α* gene (OR: 1.3, *p* < 0.00002) with POAG. This SNP lies in the 5′ UTR region of the *TNF-α* gene. None of the other SNPs in the *TNF-α* or in the 3′UTR region reached significance. We did not observe any SNPs in the *IL-2* genes as being associated with glaucoma in POAAGG cohort (Appendix A).

### 3.2. Association of Plasma TNF-α and IL2 with POAG

TNF-α expression was analyzed in plasma samples obtained from 190 subjects, 72 of which were normal subjects, and 118 POAG cases. While there was no significant difference between the gender distribution of normal subjects and POAG cases (*p* = 0.12), the mean age and BMI was significantly different between the two groups (*p* < 0.001) (Table 1).

The average concentration of the plasma TNF-α is higher in POAG cases (86.96 pg/mL) when compared to TNF-α concentration in normal subjects (60.6 pg/mL); however, the difference did not reach statistical significance (Table 1, Figure 1a). Spearman’s correlation between age and TNF-α levels is not significant, indicating that while there is a significant difference between ages in the control versus POAG case cohorts, this difference does not confound the relationship when analyzing TNF-α levels. Similarly, the Spearman’s correlation between BMI and TNF-α is also not significant, indicating that BMI is not a confounder when analyzing TNF-α levels.

IL-2 levels were analyzed in 367 African American subjects, consisting of 232 POAG cases and 135 controls. While there was no significant difference in the gender distribution between POAG cases and controls (*p* = 0.08), the mean age between the two groups was significantly different (*p* < 0.001) (Table 2).

For IL-2 analysis, the mean concentration of the plasma IL-2 was higher in controls (32.42 pg/mL) than cases (26.12 pg/mL), but the difference did not reach statistical significance (Table 2). A gender subgroup analysis showed that IL-2 levels in male cases (31.21 pg/mL) was significantly higher than in female cases (23.01 pg/mL) (*p* = 0.026) (Table 3).

Spearman correlation showed no correlation between the age of subjects and IL-2 levels in cases or controls. There was a Spearman rho of 0.04 among the 232 cases (*p* = 0.58) and a rho of 0.02 among the 135 controls (*p* = 0.75) (Table 4).

## 4. Discussion

The overall objective of our study was to test the hypothesis that immune activation is associated with glaucoma. To this end, we tested whether SNPs in *TNF-α* and *IL-2* genes are associated with g laucoma. We also measured cytokine TNF-α and IL-2 levels in the plasma of clinically well-defined POAG cases and compared them to controls to evaluate if these plasma cytokines serve as biomarkers for POAG. The practicality of aqueous humor sampling as a clinical diagnostic tool is dubious. Though previous studies have obtained aqueous humor or anterior chamber tissues from POAG patients to study differences in cytokine concentrations, this method is invasive and carries risks and complications. [23,34,35,36] Blood is easily accessible and offers a potential screening modality for diagnosing POAG. Hence, we isolated and screened plasma from blood samples collected from our POAAGG cohort. Patients with Alzheimer’s disease have differing plasma concentrations of certain signaling proteins compared to healthy subjects, suggesting that dysfunction in the central nervous system is accompanied by the systemic changes found in plasma [37]. A similar relationship could extend to POAG.

Patients with glaucoma are known to have atypical T-cell subsets and higher levels of serum antibodies against proteins in the retina and optic nerve [38]. The fact that the cellular milieu is altered in POAG implies that the immune system contributes strongly to the onset and development of optic neuropathy. Additionally, research has shown that the expression patterns of genes in the TM and Schlemm’s canal are like those of peripheral leukocytes circulating in the blood [39].

IL-2, which is regulated by Th1 cells and triggers the proliferation of T-cells, has previously been studied as a biomarker. However, it has never been investigated in an African American cohort. Prior studies have measured IL-2 levels in aqueous humor, vitreous humor, tear film, and peripheral serum samples without targeting a specific ethnicity. Serum IL-2 concentrations in POAG cases from other ethnicities were much lower than those reported in our plasma findings, which could be due to differences in ethnicity, environment, underlying comorbidities, and procurement and the processing of blood samples [20,38]. Additionally, the prior studies had smaller sample sizes (~ 50 patients), likely contributing to the lack of significant differences between groups.

Our findings did not reveal a statistically significant difference in plasma IL-2 levels between POAG cases and controls. Mean concentrations were similar among the two groups. Due to the lack of longitudinal data, it is not known how IL-2 titers change throughout the course of POAG. IL-2 in cases may be lower than in controls in the early stages of disease before increasing, or vice-versa. Additionally, we observed a mild, but significant negative correlation between IL-2 concentration and mean IOP. Although it is difficult to infer any causality, this suggests that IL-2 levels may contribute to a decreased IOP by affecting a mechanism such as outflow facility or endothelial permeability. This could also explain why IL-2 levels might be lower in POAG patients than controls. However, there is no evidence in the literature documenting such a physiology. There was no correlation between IL-2 levels and the other endophenotypic values documented in our population, such as CCT, CDR, RNFL mean thickness, or baseline MD.

While TNF-α has been studied as a biomarker in various populations, it has yet to be explicitly measured in an African American population [30,40,41]. Previous studies have measured TNF-α in the aqueous humor of POAG and normal patients without specifically measuring levels within a particular racial or ethnic group [28,29]. TNF-α is reported to be significantly higher in the plasma of POAG cases in a Saudi Arabian population. Compared to our findings, the overall TNF-α levels measured by ELISA were much lower than those reported in our findings [30], which could be due to differences in population, environment, modifying genes or other factors. It has previously been reported that increased levels of TNF-α and TNF-α-1 receptor, which result in an apoptotic response, have been found in higher quantities in retinal sections of glaucomatous eyes when compared to control donor eyes [42]. Although our results did not show a significant difference between plasma TNF-α levels in controls when compared to POAG cases, we did observe higher expression levels for plasma TNF-α in POAG cases when compared to normal subjects, which agrees with previous findings [30]. Our results may additionally be altered due to undiagnosed systemic diseases affecting TNF-α levels. The correlation between CDR, RNFL, MD, PSD, CCT and IOP and TNF-α levels did not reach significance with any tested endophenotypes. Analysis of a larger POAAGG cohort, POAG subtypes such as early and advanced POAG and a common set of diagnostic criteria for both cases and controls may provide a better understanding of the role of TNF-α and POAG. Once a better understanding of TNF-α and its effect on POAG is obtained, it would be interesting to further elucidate the mechanism by which it contributes to RGC death by focusing on the receptors that are activated by TNF-α.

The results from our study could also be influenced by underlying systemic diseases that may influence IL-2 production. Cytokines, such as TNF-α, have been shown to change based on age, infection, or other system conditions including hypertension and diabetes [43]. Patients with infectious or autoimmune disease were not excluded from our study. Our patient exclusion criteria were confounding ocular conditions. They did not include chronic inflammatory systemic illnesses such as systemic lupus erythematosus, sarcoidosis, or vasculitides. These diseases are known to upregulate inflammatory pathways, and likely impact cytokine levels.

A strength of our study was that samples were obtained from a single racial population, and this race is self-reported. However, self-reporting alone has previously been shown to be an imprecise measure of genetic ancestry. A final limitation is the lack of longitudinal data. It may be clinically relevant to record the change in TNF-α and IL-2 expression with disease progression or effective glaucoma treatment.

Although IL-2 is a major activator of T-cell proliferation, our data did not show increased levels in POAG patients. This suggests that IL-2 alterations in POAG plasma are complex and warrant further investigation. Cytokines such as IL-2 may serve a role beyond stimulating an inflammatory response. As glaucoma is on a continuum and POAG may not be just one disease but a multitude of diseases, a panel of biomarkers would be useful for categorizing the various subtypes of POAG. It is possible that plasma IL-2 levels may not be significantly altered between POAG cases and controls, but there might be a difference when the population is divided into more specific subgroups. Further studies will be helpful to explore and understand the precise mechanism of IL-2 and its pathway that is involved in the pathogenesis of POAG. Future research can also be directed toward investigating proteins that are not involved in inflammatory or oxidative stress pathways as potential biomarkers for POAG in African Americans.

Previous research has indicated specific parameters for which plasma was obtained from patients, such as at the time of cataract surgery while in a fasted state [44]. However, in our study, the plasma was not obtained at set times during the day, nor were patients in controlled satiety states. To account for these potential factors, we normalized our data to the corresponding total protein concentration for a given sample. However, we still demonstrated a large standard deviation in both the control and case populations. Alternatively, a panel of biomarkers may be needed to fully characterize the various endophenotypes of glaucoma, as this disease demonstrates a spectrum of clinical presentations. It is also possible that cytokines, IL-2 and TNF-α plasma levels may not be significantly different between cases and controls without further separating and dividing the POAG population into more specific groups. In future, other proteins, outside inflammatory and apoptotic pathways, could also be investigated as potential biomarkers to better understand this disease [45].

## Figures and Tables

**Figure 1 genes-13-00054-f001:**
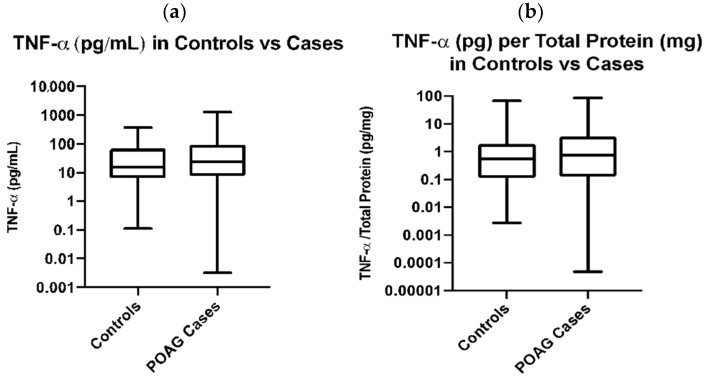
Plasma TNF-α levels in Control vs. POAG subjects. (**a**) Plasma TNF-α Levels (pg/mL) in Control vs. POAG Cases are not significantly different, determined by an unpaired *t*-test on log-transformed data; *p* = 0.68. (**b**) TNF-α levels per total protein (pg/mg) are not significantly different between cases and controls, determined by a *t*-test on log-transformed data; *p* = 0.29.

**Table 1 genes-13-00054-t001:** Demographics and TNF-α Levels in Normal vs. POAG Cases.

Variable	Normal (*n* = 72)	POAG Cases (*n* = 118)	*p*-Value
Age, mean (SD) (years)	61.82 ± 11.49	71.29 ± 12.29	<0.0001
Gender (Female)	52 (72%)	72 (61%)	0.12
Gender (Male)	20 (28%)	46 (39%)	0.31
TNF-α, mean ± SD (pg/mL)	60.6 ± 10.84	86.96 ± 15.42	0.68
TNF-α (pg/mL) per total protein (mg/mL), mean ± SD	2.47 ± 8.08	4.56 ± 10.89	0.29
BMI	32.3 ± 7.51	29.42 ± 6.12	<0.003

**Table 2 genes-13-00054-t002:** Demographics and Plasma IL-2 Levels in POAG Cases vs. Normal.

Variable	POAG Cases (*n* = 232)	Normal (*n* = 135)	*p*-Value
Age, mean (SD) (years)	69.91 ± 12.42	63.11 ± 11.52	<0.001
Gender (Female)	144 (62%)	96 (71%)	0.08
Gender (Males)	88 (38%)	39 (29%)	0.08
IL-2, mean ± SD (pg/mL)	26.12 ± 40.39	32.42 ± 61.28	0.45
IL-2 in males (pg/mL)	31.21 ± 45.51	34.92 ± 63.8	0.71
IL-2 in females (pg/mL)	23.01 ± 36.73	31.41 ± 60.53	0.96

**Table 3 genes-13-00054-t003:** Gender vs. IL-2 levels subgroup analysis.

Variable	Males	Females	*p*-Value
IL-2 in Cases (*n*) (pg/mL)	31.21 ± 45.51 (88)	23.01 ± 36.73 (144)	0.026
IL-2 in Controls (*n*) (pg/mL)	34.92 ± 63.81 (39)	31.41 ± 60.53 (96)	0.78
IL-2 in Cases + Controls (*n*) (pg/mL)	32.35 ± 51.58 (127)	26.37 ± 47.76 (240)	0.054

**Table 4 genes-13-00054-t004:** Spearman correlation of IL-2 concentrations (pg/mL) with endophenotypes.

Cases: IL-2 vs. Endophenotype	*n*	Spearman Rho	*p*-Value
Mean Intraocular Pressure	423	−0.14	0.03
Latest Central Corneal Thickness	375	0.00	0.98
Baseline Cup-to-Disc Ratio	382	0.04	0.55
Baseline Retinal Nerve Fiber Layer Mean Thickness	312	−0.02	0.80
Baseline Pattern Standard Deviation	337	−0.07	0.29
Baseline Mean Deviation	337	0.06	0.37

## Data Availability

Not applicable.

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
