# Peer review of "Evaluating TNF-α and Interleukin-2 (IL-2) Levels in African American Primary Open-Angle Glaucoma Patients"

_genes, 2021, doi:10.3390/genes13010054_

Round 1

Reviewer 1 Report

This study evaluated TNFa and IL2 for POAG in African Americans by using genetic analysis and ELISA. Unfortunately most results did not reach statistical significance. Therefore, it is not clear if this study could provide any useful information to the field. Here are my other comments:

  1. It is better to update the title into evaluating TNF and IL2 levels  in POAG patients of African Americans. If this study is to evaluate TNFa and IL2 as the POAG "biomarkers", more in-depth analyses including calculations of specificity, sensitivity, positive and negative predictive values are desirable.
  2. Please remove the full stop from the title.
  3. In the abstract, please define "AA" at its first appearance.
  4. In the abstract, results cannot fully justify the conclusion of TNFa as a "promising biomarker" for African American POAG.
  5. I do not have access to supplementary table 1. If multiple SNPs were studies, Bonferroni method should be used to correct the P values in multiple testing. 
  6. Why the number of male subjects is only shown in Table 2 but not Table 1?
  7. This study should have better exclusion criteria to avoid patients with systemic inflammatory diseases.

Author Response

This study evaluated TNFa and IL2 for POAG in African Americans by using genetic analysis and ELISA. Unfortunately, most results did not reach statistical significance. Therefore, it is not clear if this study could provide any useful information to the field. Here are my other comments:

1. It is better to update the title into evaluating TNF and IL2 levels in POAG patients of African Americans. If this study is to evaluate TNFa and IL2 as the POAG "biomarkers", more in-depth analyses including calculations of specificity, sensitivity, positive and negative predictive values are desirable.

>> We thank the reviewer for this input. As per reviewers’ suggestion, we updated the manuscript title and it now reads as, “Evaluating TNF-α and interleukin-2 (IL-2) levels in African American Primary Open-Angle Glaucoma patients”.

2. Please remove the full stop from the title.

>> We removed the period from the title as per reviewers’ suggestion.

3. In the abstract, please define "AA" at its first appearance.

>> We thank the reviewer for this suggestion and defined AA in Ln. 11 of the revised manuscript.

4. In the abstract, results cannot fully justify the conclusion of TNFa as a "promising biomarker" for African American POAG.

>> We modified the conclusion in Lns. 22-24 to now read as, “Although we found a marginal SNP association of TNF-α, assessing the expression levels of TNF-α and IL-2 may serve as promising biomarker for African American POAG.”

5. I do not have access to supplementary table 1. If multiple SNPs were studies, Bonferroni method should be used to correct the P values in multiple testing. 

>> We apologize if the Supplementary Table was not accessible during our preliminary submission. The Supplementary Table is provided with the revised manuscript for your perusal. As described in the Methods, Lns. 140-143, we determined the association of SNPs near TNF-α and IL2 including upstream and downstream regions containing 5’ and 3’ untranslated regions that were extracted from the genotyping data generated from case-control association analysis as described in POAAGG GWAS study (Ref #32). We added additional sentence from Ln. 143 to describe the methodology used to assess SNPs associated with TNF-α and IL2 genes which reads as, “Single variant, binary association tests were performed genome-wide using a logistic regression model framework as implemented in the PLATO software package and as described in POAAGG GWAS study [32].”  

6. Why is the number of male subjects only shown in Table 2 but not Table 1?

>> As per the reviewer’s suggestion, we updated the information about male subjects and included it in Table.1 of the revised manuscript.

7. This study should have better exclusion criteria to avoid patients with systemic inflammatory diseases.

>> We thank the reviewer for this suggestion. For the patients recruited in the current study, we do not have enough information to exclude patients with any systemic inflammatory diseases. This was stated as a weakness of our study in Ln. 317. However, as suggested by the reviewer, we will add systemic inflammatory diseases to the exclusion criteria for all future studies.

Reviewer 2 Report

The authors have presented the relationship between plasma IL-2, TNF-α levels with POAG and its endophenotypes, and the association of potential gene variants in IL-2 and TNF-α genes with POAG in AA population. The manuscript is written clearly and concisely. Although there are several studies testing cytokine as biomarkers in tear, aqueous humor and blood, this study combines the GWAS in Africa American, which is novel. Here are some suggestions for improvement of this article.

  1. Are the cohorts in Table 1 and 2 the same or different? Why use “Normal and POAG cases” in Table 1 and “Cases and Control” in Table 2  
  2. Figure 1, missing the label of “a”.

Author Response

The authors have presented the relationship between plasma IL-2, TNF-α levels with POAG and its endophenotypes, and the association of potential gene variants in IL-2 and TNF-α genes with POAG in AA population. The manuscript is written clearly and concisely. Although there are several studies testing cytokine as biomarkers in tear, aqueous humor and blood, this study combines the GWAS in Africa American, which is novel. Here are some suggestions for improvement of this article.

1. Are the cohorts in Table 1 and 2 the same or different? Why use “Normal and POAG cases” in Table 1 and “Cases and Control” in Table 2  

>> We thank the reviewer for pointing this out. The cohorts used in Table 1 and 2 are the same. As per reviewer’s suggestion and to improve clarity of the revised manuscript, we changed the names of Cases and Controls in Table 2 to Normal and POAG Cases.

2. Figure 1, missing the label of “a”.

>> We thank the reviewer for pointing this out. We added label a to the Figure.1 in the revised manuscript.

Round 2

Reviewer 1 Report

Thank you very much for the revision. I believe this manuscript is now suitable to be considered for publication in Genes.

This manuscript is a resubmission of an earlier submission. The following is a list of the peer review reports and author responses from that submission.